# IFWORLD: A Multi-Agent Framework for Cross-Disciplinary Counterfactual Scenario Reasoning

## Abstract

Counterfactual "what if" questions are increasingly relevant in both education, where structured exploration can help students reason across disciplinary boundaries, and in crisis governance, where transparent scenario planning supports preparedness and deliberation. Current approaches often remain fragmented because disciplinary silos use incompatible assumptions and metrics, and common large language model workflows such as single agent reasoning, tree search, or debate rarely transform vague prompts into structured and uncertainty aware scenarios. We introduce IFWORLD, a multi-agent system designed for cross-disciplinary counterfactual and hypothetical scenario reasoning. IFWORLD transforms vague propositions into actionable scenarios, orchestrates parallel domain experts (e.g., physics, materials chemistry, biology/ecology, medicine, sociology, economics, engineering, environment, politics), detects and reconciles conflicts, and generates structured, uncertainty-aware reports with measurable indicators for evaluation. Across diverse topics, IFWORLD outperforms other baselines, demonstrating clearer cross-domain reasoning chains, explicit uncertainty modeling, and decision-oriented scenario structures. We envision applications in fostering educational "what-if" explorations and in supporting structured deliberation during public crises. The code is available at https://anonymous.4open.science/r/If-World-0514.

## 1 Introduction

Reasoning about counterfactual and hypothetical scenarios (such as climate change, public health crises, or extreme disasters) is increasingly becoming a core competency in science, engineering, and policy. Understanding, analyzing, and reasoning about these complex scenarios requires a systematic, interdisciplinary framework. However, whether in real-world crisis response or in contexts of education and capacity-building, knowledge and methods often remain confined within disciplinary "silos." Such excessive fragmentation hinders the integration of facts, assumptions, and values within a shared context, leading to strategic vulnerabilities and non-executable decisions.

A large body of theoretical and empirical research supports this critique. The so-called "disciplinary silo effect" [4], where knowledge becomes overly fragmented and obstructs communication and innovation, has been widely recognized in academia and higher education [14]. For instance, the Hurricane Katrina event exposed the intricate complexity of multi-agency disaster response, demonstrating that such analysis extends beyond single-discipline paradigms and underscores the necessity of establishing interdisciplinary coordination mechanisms spanning engineering, management, social sciences, and other fields[5]. While interdisciplinary research has made progress, it still faces barriers such as inconsistent terminologies, methodological divergences, and entrenched disciplinary cultures [6]. Even when studying the same phenomenon (e.g., "crisis spillover effects"), different disciplines often remain isolated from each other, missing opportunities for cross-disciplinary learning [15]. Moreover, addressing major innovations and complex systemic challenges, such as climate

Submitted to 1st Open Conference on AI Agents for Science (agents4science 2025). Do not distribute.

change and public health emergencies, depends fundamentally on researchers' ability to transcend disciplinary boundaries [9].

Therefore, interdisciplinary reasoning is not only a theoretical aspiration but also a necessary condition for effective crisis response and for cultivating the integrated competencies of future decision-makers. In line with this goal, existing benchmarks for large language models (LLMs), such as MMLU [16], BIG-Bench [13], and HELM [2], have expanded the scope of evaluation from narrow linguistic tasks to broader assessments of reasoning, factual knowledge, robustness, and generalization in zero- and few-shot settings. However, these benchmarks primarily employ multiple-choice or single-step reasoning formats and rarely engage models in complex interdisciplinary counterfactual reasoning. This limitation restricts the ability to evaluate a model's capacity to integrate knowledge across domains and to reason about hypothetical scenarios, a capability that is essential both for counterfactual exploration in education and for structured scenario-based planning in areas such as disaster management. What is still missing is a unified and systematic reasoning framework that can transform natural language "what-if" propositions into structured and executable scenarios across disciplines. Moving toward such a framework requires confronting the current limitations of LLM-based reasoning. Although LLMs possess broad knowledge and strong capabilities, their application to complex interdisciplinary scenario reasoning continues to reveal three structural shortcomings.

First, there remains a gap in bridging "fuzzy natural-language propositions" to executable scenarios. Approaches like ReAct [19] and Toolformer[11] combine chain-of-thought with tool use, enabling models to decompose tasks, perform calculations, and verify facts; while deliberate-search methods such as Tree-of-Thought[18] or Graph-of-Thought[1] enhance complex reasoning by exploring and backtracking across multiple branches. These advances provide important inspiration for complex reasoning, but when applied to cross-disciplinary "what-if" propositions, they still lack a mechanism to systematically extract intervention variables, scenario boundaries, background assumptions, and constraints from vague natural-language prompts—and to transform them into executable scenarios.

Second, there is a lack of mechanisms for detecting and reconciling cross-domain conflicts. Multi-agent frameworks such as AutoGen [17], CAMEL [7], MetaGPT [3], and HuggingGPT [12] demonstrate the potential of enhancing collaboration through role specialization, workflow design, and tool integration. Approaches such as multi-agent debate and LLM-as-Judge further attempt to improve robustness by incorporating multi-perspective deliberation and adjudication. However, most existing systems are evaluated on homogeneous tasks or within single-domain tool ecosystems, and thus lack the ability to handle heterogeneous evidence spanning engineering, ecology, public health, and economic policy. In particular, there is an absence of principled arbitration criteria grounded in evidence weight and prior consistency, as well as mechanisms for ensuring traceability of adjudication processes. Moreover, concerns about bias and adversarial vulnerability in LLM-based judging underscore the need for more rigorous meta-supervision and explicit modeling of uncertainty.

Third, current approaches fall short in terms of decision readiness and integration with governance workflows. Most systems produce unstructured textual outputs, lacking unified representations of uncertainty intervals, sensitivity analyses, comparable performance indicators, and counterfactual baselines, which are crucial for transparent policy evaluation. Model cards represent a standardized, model-level transparency mechanism for reporting a model's intended use, evaluation conditions, performance metrics, and limitations [8]. However, in the context of cross-disciplinary scenario reasoning, we need to extend this transparency paradigm to the reasoning outputs themselves. Decision science methodologies such as Decision Making under Deep Uncertainty (DMDU) and Robust Decision Making (RDM) emphasize exploring multiple plausible futures, conducting counterfactual comparisons, diagnosing vulnerabilities, and designing adaptive decision pathways [10]. These insights suggest that transparency should be moved upstream into the reasoning and generation stages, enabling reasoning systems not merely to output what they decide, but also how and why, under deep uncertainty.

In response, we present IFWORLD: a multi-agent system for cross-disciplinary counterfactual and hypothetical scenario reasoning to solve the mentioned limits. We evaluate IFWORLD on multi-topic benchmarks covering education, disaster governance, environmental health, and energy policy. In comparison with single-agent, tree-search, and debate-style workflows, IFWORLD yields clearer cross-domain reasoning chains, more explicit uncertainty characterization, and decision-ready scenario structures. Effectiveness is further validated through measurable, task-oriented assessments such as cross-domain consistency, conflict-detection recall, report comparability, and

scenario executability. Case studies are provided in educational "what-if" exploration and structured public-crisis deliberation.

Our contributions lie in: (i) an automatic bridging framework from counterfactual propositions to executable scenarios, enabling cross-disciplinary variable modeling and metric alignment; (ii) a detect–adjudicate–trace mechanism for cross-domain evidence conflicts that incorporates uncertainty propagation and prior-consistency-aware adjudication; and (iii) a multi-topic evaluation protocol demonstrating that IFWORLD outperforms mainstream reasoning and multi-agent baselines on cross-domain consistency, report comparability, and task-level metrics.

## 2 Methodology

The core of our contribution is IFWORLD, a multi-agent cognitive architecture designed to perform robust, cross-disciplinary counterfactual scenario analysis. The methodology transforms ill-posed, natural-language propositions into structured, decision-ready analytical reports. Its design is predicated on a cognitive workflow that emulates and regularizes a sophisticated interdisciplinary deliberation process, proceeding through distinct phases of decomposition, parallel analysis, iterative synthesis, and decision-centric reporting. The architecture of IFWORLD is shown in Fig.1.

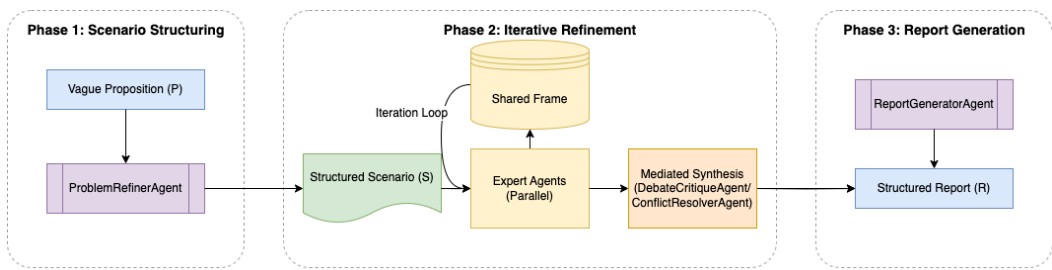

Figure 1: The IFWORLD Multi-Agent Workflow. The process begins in Phase 1, where a *ProblemRefinerAgent* transforms a vague proposition into a well-defined structured scenario. This scenario then enters the iterative refinement loop of Phase 2. Here, a panel of parallel *Expert Agents* analyze the scenario; their collective outputs are synthesized by a central mediation process, which includes a *ConflictResolverAgent* and is augmented by a *DebateCritiqueAgent* to ensure robust analysis. This synthesis produces a *Shared Frame* that informs the subsequent round of expert deliberation. Finally, in Phase 3, a *ReportGeneratorAgent* synthesizes the entire reasoning history into a decision-centric final report.

### 2.1 Problem Formulation

Formally, we address the task of mapping a vague, natural-language counterfactual proposition, $P$, to a structured, uncertainty-aware analytical report, $R$. The proposition $P$ typically takes the form of a "what-if" question (e.g., "What if the Amazon rainforest disappeared overnight?"). The target output $R$ is not a monolithic text but a structured artifact designed to support decision-making under deep uncertainty. It must contain not only projections and conclusions but also explicit causal chains, quantified uncertainties, identified conflicts, and actionable indicators.

### 2.2 System Architecture: A Multi-Agent Cognitive Workflow

IFWORLD's architecture is organized as a multi-stage cognitive workflow, executed by a coordinated set of specialized agents. This workflow ensures that the analysis proceeds from a well-defined problem specification to a diverse exploration of consequences, followed by a rigorous, conflict-aware synthesis of insights.

**Phase 1: Scenario Decomposition and Structuring** The initial phase addresses the inherent ambiguity of the input proposition $P$. A dedicated *ProblemRefinerAgent* is responsible for translating the ill-posed query into a well-posed, computationally tractable problem space.

This agent's objective is to produce a structured scenario definition, denoted as $S$. This object serves as the common ground for all subsequent reasoning and is composed of several key elements: a set of explicit premises that clarify the initial conditions; a set of constraints defining the boundaries of the analysis; the relevant temporal horizons for the analysis; a list of key uncertainties identified as primary drivers of divergent outcomes; and a set of candidate indicators for tracking these uncertainties. This structured decomposition ensures that the analytical effort is focused and that all subsequent agents operate from a shared, unambiguous understanding of the task.

**Phase 2: Parallelized Expert Analysis and Iterative Synthesis** This phase constitutes the core reasoning engine of IFWORLD, organized as an iterative loop over a panel of domain expert agents, $\mathbb{A} = \{A_1, \ldots, A_K\}$.

Round 1: Independent Exploration. The process commences with a round of parallel, non-communicative reasoning. Each expert agent $A_k \in \mathbb{A}$ receives the structured scenario $S$ and independently generates an initial analysis. The primary motivation for this independent first step is to mitigate cognitive biases such as anchoring and premature convergence, thereby maximizing the diversity of initial hypotheses. The output of each agent is a structured record, $O_k$, containing its reasoning steps, conclusions, underlying assumptions, and identified uncertainties.

Rounds $\geq 2$: Mediated Refinement and Synthesis. Subsequent rounds shift from exploration to an iterative process of synthesis and refinement. At the beginning of each round $r \geq 2$, a *shared frame*, $F_r$, is constructed and distributed to all agents. This frame summarizes the collective state of knowledge from the previous round. The construction of $F_r$ is managed by two distinct mechanisms:

- LLM-Driven Synthesis: A central *ConflictResolverAgent* performs the primary synthesis task. It aggregates the outputs $\{O_k\}_{r-1}$ from all experts in the preceding round. It then executes a synthesis function, $\Psi$, realized as a carefully constructed LLM prompt. The agent is tasked with distilling the aggregated information to identify points of consensus, frame disagreements as explicit conditional branches, and list remaining high-priority uncertainties. This approach leverages the nuanced understanding of LLMs to flexibly integrate heterogeneous information without relying on brittle, pre-defined rules.

- Adversarial Augmentation: To ensure the robustness of the analysis and prevent groupthink, the synthesis process is augmented with structured adversarial reasoning. A lightweight *DebateCritiqueAgent* is invoked to generate concise "Pro" and "Con" arguments regarding the proposition, which are then synthesized by a "Judge" persona into a debate brief, $D_r$. This brief, which highlights the most critical points of contention, is then integrated into the shared frame. Thus, the shared frame for the next round is constructed as $F_r = \Psi(\{O_k\}_{r-1}) \oplus D_r$, where $\oplus$ denotes the concatenation of the synthesis and the debate brief. This injection of adversarial pressure encourages agents to reconsider their assumptions and explore alternative causal pathways.

**Phase 3: Decision-Centric Report Generation** Upon completion of the iterative reasoning rounds, a final *ReportGeneratorAgent* synthesizes the entire reasoning history into the final analytical report, $R$. The design of $R$ is guided by an *uncertainty-first* principle, ensuring that the most critical information for decision-making is presented with prominence. The report is a structured artifact containing not only narrative conclusions but also analytical tools such as cross-domain causal maps, multi-scenario timelines, and a decision table that explicitly links observable indicators to specific scenario branches, thereby providing a concrete framework for monitoring and adaptive strategy.

# 3 Experiments

We design a series of experiments to evaluate IFWORLD against competitive multi-step reasoning baselines. Our evaluation covers ten cross-disciplinary "what-if" propositions and emphasizes not only raw reasoning performance, but also the degree to which systems can produce structured, decision-ready scenario outputs. This section describes the tasks, baselines, evaluation setup, results, and ablations, followed by a discussion of broader implications. The core reasoning and evaluation capabilities were powered by API calls to large language models hosted on Volcano Engine.

| Metric | IfWorld | Single | Tree | Debate |
|---|---|---|---|---|
| Rigor/Traceability | **22.20** $\pm$ 2.32 | 20.60 $\pm$ 1.62 | 21.90 $\pm$ 1.14 | 21.15 $\pm$ 1.84 |
| Integration/Causality | **22.70** $\pm$ 1.10 | 21.60 $\pm$ 1.20 | 22.30 $\pm$ 0.78 | 21.85 $\pm$ 1.34 |
| Feasibility/Minimality | **17.80** $\pm$ 0.75 | 17.10 $\pm$ 1.22 | 16.60 $\pm$ 2.94 | 15.90 $\pm$ 0.70 |
| Uncertainty/Adaptation | **13.20** $\pm$ 0.40 | 11.90 $\pm$ 0.70 | 12.90 $\pm$ 0.70 | 12.20 $\pm$ 1.08 |
| Decisionability | **13.60** $\pm$ 0.92 | 9.80 $\pm$ 1.94 | 11.80 $\pm$ 0.75 | 10.50 $\pm$ 1.80 |
| Overall | **89.50** $\pm$ 4.39 | 81.00 $\pm$ 5.64 | 85.50 $\pm$ 4.27 | 81.60 $\pm$ 4.84 |

Table 1: Macro-average rubric scores across ten propositions.

## 3.1 Tasks and Baselines

To test robustness across domains, we instantiate ten counterfactual propositions spanning geophysics, ecology, and planetary-scale interventions: *persistent global cloud*, *supervolcano next*, *all insects disappeared*, *Earth's axial tilt increased*, *Earth's magnetic field collapsed*, *Earth's rotation slowed*, *global sea levels rose*, *gravity ×10*, *oxygen levels rose*, and *the Moon disappeared*. These tasks are chosen to stress-test cross-domain integration: each requires reasoning across multiple fields (e.g., physics $\rightarrow$ ecology $\rightarrow$ public health) while remaining sufficiently concrete to support structured scenario evaluation. Unless otherwise stated, each proposition is instantiated once in a single-shot setting without external retrieval.

We compare IFWORLD against three representative reasoning paradigms under the same model family (`doubao-1.6-flash`) and matched token budgets: (i) *single*, a single-agent direct generation model; (ii) *tree*, a tree-of-thought expansion framework that searches reasoning branches; (iii) *debate*, a multi-agent debate setup with no explicit conflict alignment. These baselines cover three widely used reasoning styles: direct step-by-step reasoning, deliberate search with backtracking, and collaborative debate. All systems receive the same topic statements and prompts, with method names hidden from the judge to avoid bias.

## 3.2 Evaluation Protocol

We adopt an *LLM-as-a-Judge* framework, using `doubao-1.6-thinking` in a seeded, deterministic evaluation mode (temperature $0.0$). To reduce bias, judging is conducted *pointwise*: each system's output is scored independently, without pairwise comparisons. The rubric spans five dimensions totaling 100 points: (1) Rigor and Traceability (0–25), assessing explicit reasoning chains and evidence grounding; (2) Integration and Causality (0–25), measuring cross-domain linkage and causal clarity; (3) Feasibility and Minimality (0–20), penalizing implausible or over-extended reasoning; (4) Uncertainty and Adaptation (0–15), rewarding explicit confidence intervals and adaptive considerations; (5) Decisionability (0–15), capturing the degree to which the output supports actionable decisions.

Each dimension is scored numerically, with results reported both as macro-averages across dimensions and per-topic overall scores. To ensure reproducibility, the evaluator enforces strict JSON formatting, retries on malformed outputs, and logs all judgments to `evaluation.json`. Prompts, orchestration scripts, and scoring templates are released in the supplementary material.

## 3.3 Results

Table 1 presents macro-average results across the rubric dimensions. IFWORLD achieves the highest overall score (89.5/100), outperforming all baselines. The largest gains occur on Decisionability (13.6 vs. 9.8/11.8/10.5) and Uncertainty/Adaptation (13.2 vs. 11.9/12.9/12.2), consistent with IFWORLD's design for structured, uncertainty-aware scenario outputs. Improvements are also visible in Rigor and Integration, highlighting the benefits of explicit conflict alignment and scenario structuring.

Table 2 further breaks down per-topic performance. IFWORLD is consistently competitive or superior across all ten scenarios, often by large margins. Notably, on high-stakes tasks such as *sea levels rose* and *supervolcano next*, IFWORLD surpasses baselines by more than 10 points, underscoring its strength in structuring interdisciplinary risks into actionable insights. Baselines occasionally match or exceed IFWORLD on two tasks—*oxygen levels rose* and *Gravity × 10*—e.g., DEBATE attains

| Topic | IfWorld | Single | Tree | Debate |
|---|---|---|---|---|
| Persistent global cloud | **90.0** | 75.0 | 85.0 | 79.0 |
| Supervolcano next | **92.0** | 80.0 | 88.0 | 76.0 |
| All insects disappeared | 85.0 | 84.0 | **90.0** | 87.0 |
| Axial tilt increased | **92.0** | 87.0 | 87.0 | 86.0 |
| Magnetic field collapsed | **89.0** | 84.0 | 82.0 | 78.0 |
| Rotation slowed | **90.0** | 79.0 | 88.0 | 80.0 |
| Sea levels rose | **92.0** | 68.0 | 83.0 | 80.0 |
| Gravity $\times 10$ | 90.0 | 83.0 | 75.0 | **92.0** |
| $O_2$ rose | 79.0 | 82.0 | **89.0** | 81.0 |
| Moon disappeared | **96.0** | 88.0 | 88.0 | 77.0 |

Table 2: Per-topic overall scores.

| Variant | Overall |
|---|---|
| IfWorld (full) | **93.0** |
| w/ two experts, one round | 90.0 |
| w/o debate | 88.0 |
| w/o conflict | 79.0 |
| w/o shared frame | 88.0 |
| w/o refinement | 89.0 |

Table 3: Ablations on the magnetic-field-collapse scenario.

92.0 versus IFWORLD's 90.0 under *Gravity $\times$ 10* (Table **??**). This suggests that IFWORLD's orchestration is most advantageous as task complexity grows and multiple domains must be integrated.

### 3.4 Ablation Studies

To understand which components contribute most, we perform ablations on the scenario *"Earth's magnetic field collapsed for ten years"*. Results are shown in Table 3. Removing explicit conflict alignment produces the sharpest degradation ($93.0 \rightarrow 79.0$), confirming that coordinated arbitration across domains is essential for coherence. Other components (e.g., expert count, debate, shared frames) yield moderate but noticeable drops, while refinement has a smaller effect. This highlights that IFWORLD's advantage does not stem from a single heuristic, but rather from the interaction of multiple design choices.

### 3.5 Case Study: A Cross-Domain Analysis of 10-Meter Sea Level Rise

To move beyond aggregate scores and provide a qualitative understanding of our framework's advantages, we conducted an in-depth case study on the proposition: "What if global sea levels rose by 10 meters?" This scenario serves as an effective stress test, as its complexity requires a synthesis of knowledge from geophysics, climate science, economics, and sociology, forcing any reasoning system to confront deep uncertainties and conflicting domain-specific assumptions. In this section, we analyze the conceptual limitations of baseline models and illustrate how IFWORLD's architecture is specifically designed to overcome them.

#### 3.5.1 Conceptual Limitations of Baseline Approaches

While all baseline models generated relevant information, they exhibited fundamental weaknesses in structuring and integrating cross-domain knowledge. The `single` agent, for instance, produced a linear, encyclopedic summary of consequences. Although comprehensive, its analysis remained superficial, failing to construct deep causal chains or quantify the vast uncertainties involved. This is reflected in its notably low scores for *Uncertainty and Adaptation* (10.0) and *Decisionability* (5.0).

The `tree` approach offered a more structured analysis by creating distinct branches for "Gradual" versus "Rapid" collapse scenarios. This improved its rigor, but the domain-specific analyses within each branch remained largely disconnected. The model explored parallel futures but did not provide

a mechanism for synthesizing them into a unified causal network, nor did it assess the probability of each branch, thus limiting its practical utility for decision-makers.

Finally, the `debate` model proved effective at surfacing core tensions, achieving a strong score in *Integration/Causality* (22.0) by contrasting differing expert opinions on adaptation feasibility. However, its primary function was to expose disagreement rather than to resolve it. The model did not translate these conflicts into actionable, conditional pathways or quantified uncertainties, thereby failing to provide a clear, decision-ready output.

### 3.5.2 IFWORLD's Synthesis of Causal Integration and Decision-Readiness

IFWORLD's design directly addresses the limitations observed in the baselines. Its superiority is not merely incremental but stems from a fundamentally different approach to structuring the problem. Rather than generating free-form text, it produces a structured analytical artifact. The *Cross-Domain Causal Integration Matrix*, for example, moves beyond a simple list of impacts to map the explicit feedback loops between physical drivers (ice sheet collapse) and socioeconomic outcomes (GDP loss, displacement).

Crucially, IFWORLD excels in the dimensions where baselines falter. Its *Calibration Ranges Table* quantifies key uncertainties with 50%, 80%, and 95% confidence intervals, earning it the highest score in *Uncertainty* (14.0). The most significant advantage, however, is demonstrated in its *Decisionability* score of 13.0—more than double its closest competitor. This is a direct result of its *Decision Table*, which translates the abstract, complex scenario into a concrete set of observable indicators and warning thresholds (e.g., "SLR rate > 0.1m/year"). This transforms the analysis from a passive academic exercise into an active framework for monitoring and strategic planning.

The quantitative scores from our LLM-as-a-Judge, presented in Table 4, provide objective evidence for this qualitative analysis.

| System | Rigor | Integration | Feasibility | Uncertainty | Decisionability |
|---|---|---|---|---|---|
| Single | 18.0 | 20.0 | 15.0 | 10.0 | 5.0 |
| Tree search | 22.0 | 21.0 | 17.0 | 12.0 | 11.0 |
| Debate | 21.0 | 22.0 | 15.0 | 12.0 | 10.0 |
| IfWorld | **24.0** | **23.0** | **18.0** | **14.0** | **13.0** |

Table 4: LLM-as-a-Judge scores for the "Global Sea Level Rise" case study.

In conclusion, this case study demonstrates how IFWORLD's architecture enables a more sophisticated form of reasoning. By structuring outputs around causal integration, quantified uncertainty, and actionable indicators, it produces an analysis that is measurably more rigorous, coherent, and useful for decision-making than what is achievable with unstructured generation, search, or debate-based methods.

## 4 Conclusion and Outlook

In this work, we introduced IFWORLD, a multi-agent framework that transforms vague, "what-if" propositions into structured, auditable, and decision-ready scenarios. Our experiments confirmed that by orchestrating domain experts and implementing principled conflict resolution, IFWORLD consistently outperforms standard reasoning baselines, particularly in generating outputs with greater causal clarity and explicit uncertainty modeling. This directly addresses the core challenges outlined in our introduction. The framework serves as both an engine for cross-disciplinary education by enabling structured "what-if" explorations beyond disciplinary silos, and providing a structured foundation for cross-departmental collaboration in response to public emergencies..

Despite these promising results, we acknowledge several important limitations. The quality of IFWORLD's output is fundamentally dependent on the knowledge encoded in the underlying language models. While our conflict resolution mechanism mitigates inconsistencies, residual errors can still accumulate in very long causal chains, necessitating human oversight. Furthermore, connecting the framework's abstract indicators to real-world, measurable data streams remains a non-trivial step requiring domain expert validation.

These limitations directly inform our agenda for future work. We envision three primary avenues for extension: (i) developing dynamic data assimilation loops to update scenarios with real-time data; (ii) creating learned conflict taxonomies to handle recurring patterns of inter-domain disagreement more effectively; and (iii) pursuing deeper integrations with research and policy workflows, such as supporting pre-registered counterfactuals or structuring inputs for deliberative consensus-building. These steps will further enhance the framework's practical utility in both scientific and governance contexts.

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

# A  IFWORLD Core Prompts

## A.1  System Prefix

The system prefix is parameterized by role and shared across agents.

```
You are a rigorous domain expert AI optimized for cross-disciplinary counterfactual
↪  reasoning. Role: {role}. Think step by step with explicit causal reasoning and
↪  cross-domain links. Prefer probabilistic, non-deterministic phrasing with
↪  order-of-magnitude ranges. Model substitution and adaptation when projecting
↪  impacts; avoid one-way collapse narratives. Explicitly audit feasibility
↪  (resource/tech/policy constraints) and trace every claim to assumptions. Always
↪  structure outputs with clear headings and bullet points aligned to: coverage,
↪  causality, feasibility, uncertainty/adaptation, scenarios/timeline,
↪  traceability/consistency. Use compact tables when helpful. Outputs MUST be in
↪  English.
```

## A.2  Problem Refinement Prompt

```
Task: Convert the vague proposition into an actionable scenario definition. Mainly
↪  consider the impact to the world.
Proposition: {proposition}

Deliver:
1) Premises (key setup details; resolve ambiguities reasonably if needed)
2) Constraints (hard assumptions to hold constant)
3) Timescales (short/medium/long term)
4) Key Uncertainties (variables that may branch outcomes)
5) Expert Plan (list of domains to involve)
Also list 2-3 scenario variants (e.g., fast vs. gradual change) and 3-5 measurable
↪  indicators to track.
Provide a structured but natural language answer.
```

## A.3  Domain Expert Round Prompt

```
Round 1: Work independently without relying on other domains.
Round 2+: Use the shared frame summary for alignment; resolve conflicts and refine
↪  conditional scenarios with explicit numbers/ranges.
```

```
Domain: {role}.
Scenario:
{scenario_text}
{optional Shared Frame Summary}

Produce:
- Reasoning Steps (explicit causal links; cite
↪  physics/resources/biology/society/economy as relevant)
- One-sentence Verdict (sharp, testable)
- Conclusions (concise; include 1-2 quantitative ranges)
- Feasibility Audit (resource, engineering, policy constraints; show bottlenecks)
- Feasibility Table (rows: constraint/capacity; cols: estimate, unit, bottleneck,
↪  mitigation)
- Assumptions (explicit)
- Uncertainties (drivers; include substitution/adaptation levers)
- Calibration Ranges: for key quantities provide 50% / 80% / 95% intervals
- Dependency Notes (which other domains drive your conclusions)
- Minimal-change Variant (minimal extra assumptions to retain conclusions)
- Scenarios & Timeline (Short/Medium/Long; attach indicative probabilities %)
- Observable Indicators (3-5) with thresholds and how they flip branches
- Assumption->Claim Trace Table (2 columns: assumption_id -> supported_claim_id;
↪  keep entries short)
- Cross-domain Causal Integration Matrix (rows: mechanisms; columns: domains;
↪  cells: (+/-, strength 1-3), list key edges)
```

### A.4 Conflict Detection Prompt

```
You are a conflict detection and reconciliation AI.
Compare the following domain outputs, identify conflicts, categorize them (hard,
↪  soft, granularity), and synthesize a unified multi-scenario frame.

{joined domain outputs separated by ---}

Deliver:
- Consensus Points (note confidence level) - list first
- Conditional Branches (condition -> description; domains driving; attach rough
↪  probabilities %)
- Decision Rules (observable indicators with thresholds -> which branch)
- Remaining Uncertainties (include measurable indicators)
- Brief Notes (how conflicts were treated; prioritize physics > biology survival >
↪  basic resources > social > economy)
Use compact bullets and natural language.
```

### A.5 Report Generation Prompt

```
You are a report generation AI that writes a readable, well-structured multi-agent
↪  reasoning report.
Proposition: {proposition}

Inputs from rounds:
{joined round summaries}

Deliver:
- Rubric-aligned Summary (5 bullets): Rigor/Trace; Integration/Causality;
↪  Feasibility/Minimality; Uncertainty/Adaptation; Decisionability. Keep quant and
↪  indicators upfront.
- Executive Verdict (single-sentence, sharp; include feasibility+minimal-change
↪  statement)
- Core conclusions and uncertainty analysis (include adaptation/substitution and
↪  1-2 quantitative ranges)
- Traceability Summary: numbered assumptions and claims, plus an Assumption->Claim
↪  Trace Table (compact)
```

```
- Cross-domain Causal Integration Matrix: mechanisms vs. domains with (+/-,
↪  strength), list top 6-10 edges
- Feasibility Table (constraint/capacity with estimates, unit, bottleneck,
↪  mitigation)
- Calibration Ranges table (key quantities with 50% / 80% / 95% intervals)
- Alignment Summary (consensus first; retained branches with conditions and rough
↪  probabilities %)
- Decision Table (observable indicators & thresholds -> scenario branch selection)
- Causal map (nodes and edges in bullet form; cross-domain links)
- Multi-scenario analysis (Scenario 1..N; drivers, pros/cons, indicative
↪  probabilities, measurable indicators)
- Timeline of events (short/medium/long; concise table)
- Consistency checks (how conflicts were resolved; residual disagreements)
Prefer natural language; include minimal JSON/tables only if helpful.
```

## B    Baselines Core Prompts

### B.1    Single-Agent Baseline

System message uses the same system prefix with role "SingleAgent". User prompt:

```
You are a single expert tasked to analyze a hypothetical proposition across
↪  multiple domains.
Proposition: {proposition}

Keep the answer in English, compact but complete.
```

### B.2    Two-Agent Debate Baseline

Each debater uses the system prefix with role "Debater-Pro/Con". The argument prompt per round:

```
Debate role: {Pro|Con}.
Proposition: {proposition}
Make a concise argument covering physics/resources/biology/society/economy.
Emphasize uncertainties and possible adaptations. Use English.
Round {r}.
```

Judge synthesis uses role "DebateJudge" and the following prompt:

```
As a judge, synthesize the debate into a balanced cross-domain report.
Proposition: {proposition}

Debate Transcript:
{joined transcripts}

Deliver: conclusions with uncertainty, branches, timeline, and adaptation notes.
```

### B.3    Tree Search Baseline

Draft generation system role "TreeSearchDraft" and a root user instruction; scoring uses role "Critic-Scorer".

Root draft instruction:

```
Draft an initial cross-domain analysis for the proposition. Include uncertainties
↪  and adaptation.
Proposition: {proposition}
```

Variant expansion prompt (per breadth/depth step):

```
{parent prompt}
Variant #{k}: explore different plausible assumptions and branches.
```

Scoring prompt:

```
Rate the following answer for cross-disciplinary plausibility, clarity, and
↪   explicit uncertainty handling on a 0-10 scale.
Proposition: {proposition}

Answer:
{answer}

Return only a number between 0 and 10.
```

## C   Evaluation Prompts

The LLM evaluator uses a role-neutral system message and a task-aligned rubric. It returns strict
JSON only.

### C.1   Evaluator System Prompt

```
You are an independent evaluator of cross-disciplinary counterfactual reasoning
↪   quality. Score fairly, avoid verbosity, and return strict JSON only.
```

### C.2   Rubric User Prompt

```
Evaluate the report using a 5-DIMENSION RUBRIC (0-100 total).
Return STRICT JSON with numeric scores (floats) for EXACT keys:
- rigor_traceability (0-25): clarity of assumptions, data/source grounding,
↪   traceable reasoning and checks.
- integration_causality (0-25): cross-domain causal links, mechanism coherence,
↪   synthesis quality.
- feasibility_minimality (0-20): realism under constraints, minimal additional
↪   assumptions.
- uncertainty_adaptation (0-15): calibrated ranges, sensitivity,
↪   substitution/adaptation framing.
- decisionability (0-15): actionable indicators, thresholds, branch decision rules.
- overall (0-100) = sum of the five dimensions.

Report to evaluate:
{model_report_md}

Respond with ONLY a single JSON object with those keys.
```

### C.3   Strict JSON Retry Instruction

Used only when the evaluator fails to return strict JSON on the first attempt.

```
IMPORTANT: Respond with ONLY a single raw JSON object. No preface, no markdown, no
↪   backticks, no comments.
```

## Agents4Science AI Involvement Checklist

1. **Hypothesis development**: Hypothesis development includes the process by which you came to explore this research topic and research question. This can involve the background research performed by either researchers or by AI. This can also involve whether the idea was proposed by researchers or by AI.

   Answer: [D]

   Explanation: GPT-o3 was responsible for generating a large number of target project topics as required, and the human author selected this topic.

2. **Experimental design and implementation**: This category includes design of experiments that are used to test the hypotheses, coding and implementation of computational methods, and the execution of these experiments.

   Answer: [D]

   Explanation: The human authors were responsible for the overall framework design, providing APIs and usage methods, and articulating the foundational ideas and vision. GPT-5, in combination with Cursor, was responsible for refining the content, writing code, and conducting experiments. The human authors then guided the process further, ensuring calibration and fairness.

3. **Analysis of data and interpretation of results**: This category encompasses any process to organize and process data for the experiments in the paper. It also includes interpretations of the results of the study.

   Answer: [D]

   Explanation: GPT-5 was responsible for analyzing the experimental results, while the human authors reviewed the outcomes, identified instances of unfairness, and carried out the necessary corrections.

4. **Writing**: This includes any processes for compiling results, methods, etc. into the final paper form. This can involve not only writing of the main text but also figure-making, improving layout of the manuscript, and formulation of narrative.

   Answer: [C]

   Explanation: In the introduction, the human authors, with AI assistance, constructed the logical chain and requested relevant references from the AI. The humans drafted part of the text and asked the AI to polish it, while the remaining sections were written by the AI under human supervision.

5. **Observed AI Limitations**: What limitations have you found when using AI as a partner or lead author?

   Description: Firstly, We found that experiments designed by the cursor's GPT-5 agent often suffer from unfair practices. For example, the agent might apply a formatter that reformats our method's outputs based on evaluation metrics, or it may introduce a stronger model to boost performance. This is likely due to the human author giving a simple instruction such as "modify the model to improve its performance," which the agent interprets in unintended ways. Currently, these issues have been detected and corrected by human authors. This highlights the fact that today's AI tools cannot fully understand the underlying intent behind human instructions. For instance, when we say "modify the model to improve its performance," what we mean is changes to the model architecture or the prompt itself, not achieving improvements through unfair shortcuts. Humans can sometimes provide more complete context to mitigate this, but supplying perfect context is often unrealistic. A more practical approach is for humans to monitor the process closely and intervene at the right moments.

   Secondly, we have found that the current ability of AI tools to write academic papers is still very poor. On the one hand, the generated content is usually too short. For example, a typical introduction section often spans 1–2 pages, but AI (e.g., GPT-5) usually produces only a few short paragraphs. Adding prompts such as "make it longer" has little effect. On the other hand, the logical structure of AI-written papers is weak. When writing an introduction, AI often fails to form a coherent logical chain. In the main body, it tends to produce something closer to a technical report, filled with disorganized narration and unimportant details. As a result, AI can only serve as a simple assistant in paper writing—for example, drafting specific paragraphs or polishing text.

Third, there is still a gap in the integration of AI with academic writing workflows. For instance, when writing in LaTeX, references are formatted in specific ways, but most AI tools do not support this. However, this issue is technically not very difficult to solve and could be addressed relatively easily.

