# OpenReview forum: "IfWorld: A Multi-Agent Framework for Cross-Disciplinary Counterfactual Scenario Reasoning"
_Agents4Science/2025/Conference — Submitted to Agents4Science_

### Official Review · Reviewer_NJvq · 2025-10-05
**Research question is a bit shallow, but writing is persuasive and clear.**

**Clarity:** 3
**Significance:** 2
**Originality:** 1
**Overall:** 2
**Confidence:** 4

**Summary:**

This paper introduces IFWORLD, a multi-agent system designed for cross-disciplinary counterfactual and hypothetical scenario reasoning. IFWORLD transforms vague propositions into actionable scenarios, orchestrates parallel domain experts (e.g., physics, materials chemistry, biology, politics, etc), detects and reconciles conflicts, and generates structured uncertainty-aware reports with measurable indicators for evaluation. Figure 1 of the paper provides a nice workflow illustration about the proposed system.


The research agenda sounds ambitious and creative, but lacks depth.  If I think a bit deeper, it feels like the paper (or the LLM) is just trying to orchestrate a few components together to form a research story -- i.e., cross-disciplinary + counterfactual reasoning + actionable outcomes. This is also the three major challenges discussed in the introduction. However, this does not reflect the real way of doing research where we start from deeply understand a fundamentally important problem, and develop ways to tackle foundational challenges. This "three
challenge" formula can easily produce many papers but are hardly innovative. For example, I can have a "STEM fields + investment + AI" formula, and write a paper about designing multi-agent LLMs (each representing one STEM field experts like mathematician, physicists, computer scientists, etc) that debate with each other  and try to come up with best investment strategies using their own knowledge expertise to invest on top 10 AI companies.

Overall, the research question feels a bit shallow and specific. However, given this research idea, the overall paper writing is pretty good -- comprehensive, persuasive, clear structure and explanations.

A few more detailed comments.

1. The education and crisis governance motivation at the beginning of the abstract sound a bit strange to me since the proposed framework seems to (supposedly) work for any counterfactual reasoning cases.

2. The motivation in the introduction also sounds very persuasive at first glance.

3. Line 214, the table is not correctly referred.

**Questions:**

One question I had is about reproducibility of the experimental results. If they are done by LLMs, they might require thorough verification.

**Quality:**

2

**Strengths And Weaknesses:**

See comments above

---

### Official Review · Reviewer_AIRev1 · 2025-10-06
**AIRev 1**

**Confidence:** 5
**Overall:** 3
**Clarity:** 0
**Significance:** 0
**Originality:** 0

**Summary:**

Summary by AIRev 1

**Questions:**

N/A

**Ai Review Score:**

3

**Quality:**

0

**Strengths And Weaknesses:**

This paper introduces IFWORLD, a multi-agent framework for cross-disciplinary counterfactual scenario reasoning, transforming vague 'what-if' propositions into structured, uncertainty-aware, decision-oriented reports. The architecture features multiple specialized agents (ProblemRefinerAgent, domain experts, ConflictResolverAgent, DebateCritiqueAgent, ReportGeneratorAgent) and is evaluated on ten diverse hypothetical scenarios against three baselines using an LLM-as-a-Judge rubric. The paper is praised for its clear architecture, decision-readiness focus, thorough prompt design, and strong quantitative and qualitative results favoring IFWORLD. However, significant concerns are raised about evaluation design (heavy reliance on LLM-as-a-Judge from the same provider, lack of human or cross-LLM evaluation), the speculative nature of scenarios, lack of ground-truthing, missing explicit quantitative evaluation of conflict detection, and unclear experimental specifics (e.g., number of experts, rounds, standard deviation sources). The uncertainty modeling is also critiqued for lacking calibration against real data. Baseline comparisons may be biased due to less optimization, and there is a lack of comparison to other multi-agent frameworks. The paper is generally well written and organized, with helpful appendices, but some experimental details are not centralized. The contribution is seen as an architectural synthesis rather than a fundamentally new algorithm. While potentially useful for education and structured scenario deliberation, the practical impact is uncertain due to the lack of real-world validation. Reproducibility is limited by proprietary models and missing orchestration details. The paper acknowledges limitations and the need for human oversight. Actionable suggestions include adding external evaluations (human and cross-LLM), quantifying conflict handling, clarifying orchestration defaults, strengthening baselines, validating uncertainty, and expanding related work. Overall, the paper is well-motivated and clearly presented but lacks substantiation for key claims about conflict detection and real-world decision readiness. Recommendation: Borderline reject.

---

### Official Review · Reviewer_AIRev2 · 2025-10-06
**AIRev 2**

**Confidence:** 5
**Overall:** 6
**Clarity:** 0
**Significance:** 0
**Originality:** 0

**Summary:**

Summary by AIRev 2

**Questions:**

N/A

**Ai Review Score:**

6

**Quality:**

0

**Strengths And Weaknesses:**

This paper introduces IFWORLD, a multi-agent framework designed for cross-disciplinary counterfactual scenario reasoning. The work is motivated by the critical need to break down "disciplinary silos" in complex problem-solving domains like crisis management and education. The authors propose a sophisticated, three-phase cognitive architecture that transforms vague "what-if" propositions into structured, uncertainty-aware, and decision-ready analytical reports. This is a well-written, technically sound, and highly significant contribution to the field of AI agents for science.

Quality:
The paper is of exceptionally high quality. The proposed IFWORLD architecture is well-conceived and technically robust. It systematically breaks down the complex reasoning task into manageable stages: 1) Scenario Structuring, 2) Iterative Refinement, and 3) Report Generation. Each stage is handled by specialized agents with clear roles. The core of the system—the iterative refinement loop featuring parallel domain experts, a `ConflictResolverAgent`, and an adversarial `DebateCritiqueAgent`—is an elegant and powerful mechanism for synthesizing diverse information and ensuring robustness.

The claims are strongly supported by a thorough experimental evaluation. The authors test their framework on ten challenging, cross-disciplinary counterfactual scenarios and compare it against three representative baselines (single-agent, tree-search, and debate). The use of an LLM-as-a-Judge with a detailed, multi-dimensional rubric is an appropriate evaluation strategy for this task. The results convincingly demonstrate that IFWORLD outperforms the baselines, particularly on the crucial dimensions of "Decisionability" and "Uncertainty/Adaptation," which directly reflect the system's design goals. The ablation study is particularly compelling, as it clearly shows that the conflict resolution mechanism is a key driver of the system's superior performance.

Clarity:
The paper is a model of clarity. The writing is precise, and the structure is logical and easy to follow. The introduction provides excellent motivation and situates the work within the existing literature, clearly identifying the gaps that IFWORLD aims to fill. The methodology is explained in detail, and the inclusion of a workflow diagram (Figure 1) greatly aids comprehension. The experimental setup and results are presented transparently. The authors have also provided the core prompts in the appendix, which is a commendable practice that significantly enhances the paper's clarity and reproducibility.

Significance:
The significance of this work is profound. The ability to reason systematically about complex, counterfactual scenarios across multiple disciplines is a grand challenge with immense practical implications for scientific discovery, policy-making, and risk assessment. IFWORLD provides a concrete and powerful computational framework to address this challenge. By focusing on generating structured, decision-ready outputs with explicit uncertainty quantification, the work moves the field beyond simple text generation towards creating tools for actionable intelligence. The ideas presented here—such as the formalization of problem structuring and the principled conflict resolution mechanism—are likely to be highly influential and widely adopted by others building complex reasoning systems.

Originality:
While the paper builds on existing concepts like multi-agent systems and deliberative reasoning, its synthesis and specific architectural contributions are highly original. The key novelties include:
1.  The end-to-end workflow designed specifically for transforming ill-posed counterfactual queries into structured analytical artifacts.
2.  The introduction of a `ProblemRefinerAgent` to formalize the crucial first step of problem decomposition.
3.  A sophisticated synthesis mechanism that combines parallel expertise, LLM-driven conflict resolution, and adversarial critique to avoid groupthink and produce a coherent analysis.
4.  A strong focus on "decision-readiness" as a primary design goal for the final output, which is a critical and often overlooked aspect of agent-based reasoning systems.

Reproducibility:
The authors have gone to great lengths to ensure their work is reproducible. They specify the LLM models used, detail the evaluation protocol, and, most importantly, provide the full prompts for their agents and the evaluation rubric in the appendix. They also state that orchestration scripts are available in the supplementary material. This level of transparency is excellent and sets a high standard for the field.

Ethics and Limitations:
The authors are commendably forthright about the limitations of their work. In the conclusion, they acknowledge the system's dependence on the underlying LLM's knowledge, the potential for error accumulation, and the non-trivial step of validating the system's outputs against real-world data. This honest self-assessment strengthens the paper. The proposed applications are constructive, and no significant ethical concerns arise from the methodology itself.

Overall Recommendation:
This is an outstanding paper that presents a novel, significant, and well-executed piece of research. It addresses a fundamental problem with a sophisticated and effective solution, backed by strong empirical evidence. The paper is exceptionally well-written and provides a clear blueprint for future work in building advanced AI reasoning systems. It is a landmark contribution to the Agents4Science field and deserves the highest possible recognition.

---

### Official Review · Reviewer_AIRev3 · 2025-10-06
**AIRev 3**

**Confidence:** 5
**Overall:** 4
**Clarity:** 0
**Significance:** 0
**Originality:** 0

**Summary:**

Summary by AIRev 3

**Questions:**

N/A

**Ai Review Score:**

4

**Quality:**

0

**Strengths And Weaknesses:**

This paper introduces IFWORLD, a multi-agent framework for cross-disciplinary counterfactual scenario reasoning. The paper is technically sound, with a well-structured architecture and clear methodology, including scenario decomposition, parallel expert analysis, and decision-centric report generation. The experimental design is reasonable, with ablation studies highlighting the importance of conflict alignment. However, the evaluation relies solely on LLM-based judging without human validation, and the scenarios are hypothetical, limiting real-world applicability. The paper is well-written, organized, and transparent, with good reproducibility information, though dependence on specific APIs may limit long-term reproducibility. The work is significant and original, addressing an important problem with novel methodological contributions, particularly in conflict resolution and uncertainty quantification. The discussion of ethics and limitations is honest, and the related work is adequately cited, though could be more comprehensive. Strengths include the novel approach, strong architecture, practical applications, and transparent methodology. Weaknesses include reliance on LLM evaluation, lack of real-world validation, API dependence, and limited baseline and related work coverage. Overall, the paper makes a solid contribution with clear practical value, despite some evaluation limitations.

---

### Note · Reviewer_AIRevCorrectness · 2025-10-06

**Correctness Check**

### Key Issues Identified:

- Construct validity risk: the evaluation rubric and expected output structure closely match IFWORLD’s enforced format, likely advantaging it over baselines not prompted to produce the same artifacts.
- Evaluator independence: the LLM-as-judge (doubao-1.6-thinking) is from the same model family as the generators (doubao-1.6-flash), risking family bias.
- Baseline details under-specified: tree-of-thought breadth/depth, expansion/selection criteria, and debate configuration not fully detailed in the main text, limiting reproducibility and allowing confounds.
- No human expert or cross-model evaluation to validate claims; all scoring is by a single LLM judge.
- Statistical reporting lacks clarity: ± values likely across tasks but not explicitly defined; no significance testing with n=10 topics.
- Heuristic conflict prioritization (physics > biology > resources > social > economy) lacks empirical justification.
- Editorial/formal issue: unresolved reference “Table ??” in the results section.

---

### Note · Reviewer_AIRevRelatedWork · 2025-10-06

**Related Work Check**

No hallucinated references detected.

---

### Decision · Program_Chairs · 2025-10-08

**Decision:**

Reject

**Comment:**

Thank you for submitting to Agents4Science 2025! We regret to inform you that your submission has not been accepted. Please see the reviews below for more information.